# Model Autophagy Analysis to Explicate Self-consumption within Human-AI Interactions

**Shu Yang**[*,1,2,3], **Muhammad Asif Ali**[*,1,2], **Lu Yu**[5], **Lijie Hu**[†,1,2,4], and **Di Wang**[†,1,2,4]

[1]Provable Responsible AI and Data Analytics (PRADA) Lab
[2]King Abdullah University of Science and Technology
[3]University of Macau    [4]SDAIA-KAUST AI    [5]Ant Group

## Abstract

The increasing significance of large models and their multi-modal variants in societal information processing has ignited debates on social safety and ethics. However, there exists a paucity of comprehensive analysis for: (i) the interactions between human and artificial intelligence systems, and (ii) understanding and addressing the associated limitations. To bridge this gap, we present Model Autophagy Analysis for large models' self-consumption explanation. We employ two distinct autophagous loops (referred to as "self-consumption loops") to elucidate the suppression of human-generated information in the exchange between human and AI systems. Through comprehensive experiments on diverse datasets, we evaluate the capacities of generated models as both creators and disseminators of information. Our key findings reveal: (i) A progressive prevalence of model-generated synthetic information over time within training datasets compared to human-generated information; (ii) The discernible tendency of large models, when acting as information transmitters across multiple iterations, to selectively modify or prioritize specific contents; and (iii) The potential for a reduction in the diversity of socially or human-generated information, leading to bottlenecks in the performance enhancement of large models and confining them to local optima.

## 1 Introduction

Large models, including large language models (LLMs) (Bai et al., 2022b; Zeng et al., 2022; OpenAI, 2023; Touvron et al., 2023) and large multi-modal models (Yang et al., 2023; Yin et al., 2023), are rapidly emerging as transformative tools, reshaping our world in a formidable way. Among their myriad implications, their growing social impact stands out, making them an integral component of our modern communication era. These models facilitate the dissemination of viewpoints and information within human society by engaging in continual interaction with humans (Gao et al., 2023; Bian et al., 2023). Particularly noteworthy are recent technological advancements that have sparked an arms race, resulting in the daily training of hundreds of next-generation models using a blend of real (human-generated) and synthetic (LLM-generated) data. This iterative training process engenders an autophagous loop (elucidated in Section 3.1) within the datasets, wherein new models are continually trained on synthetic data. Previous investigations by Alemohammad et al. (2023), and Shumailov et al. (2023b) reported the decline in data quality and diversity with repeatedly generated data, often employed to train visual generative models, a phenomenon termed Model Autophagy Disorder. They underscored the dearth of fresh and realistic training data as a primary driver of this disorder. However, their analyses entirely relied on simulated experiments to demonstrate the decline in model performance. Therefore, a deeper examination is warranted to elucidate why real data is becoming increasingly scarce and its implications for the flow of information in human society. We argue that despite their status as novel and consequential components of the communication era (Edwards et al.,

---

[*]The first two authors contributed equally to this work.
[†]Corresponding author.

2016), the inherent limitations of large models remain inadequately explored. Specifically, we aim to answer the following questions: (i) What impacts do human-generated real and synthetic data have on model training? (ii) To what extent do samples from repeatedly generated synthetic data influence data quality versus diversity? (iii) What social ramifications will repeated data loops have on information dissemination?

To bridge this gap, in this research, we present Model Autophagy Analysis for explicating self-consumption within large models. There exist several **motivations** underlying this work. *Firstly*, large models are extensively being utilized across various domains (Kaddour et al., 2023), and even crowd-sourced annotators heavily rely on generative AI for data curation and decision-making (Veselovsky et al., 2023). *Secondly*, with internet being a direct source of training data, the contemporary models are unwittingly being trained on AI-synthesized data (Alemohammad et al., 2023; Shumailov et al., 2023a; Veselovsky et al., 2023). *Thirdly*, numerous studies opt to use the models as the generators and selectors of their training data, aiming to reduce overall training costs (Li et al., 2023; Huang et al., 2023a). This trend strictly implies that emerging large models are predominantly being trained on synthetic data, subsequently shaping subsequent human endeavors upon this synthetic foundation. We term this phenomenon as autophagy ("self-consumption").

Our work employs the concept of autophagous loops to analyze and comprehend the flow of information. Specifically, it introduces two distinct variants of autophagous loops based on how humans and large models construct and utilize data or information from their surroundings, as illustrated in Figure 1 and 2, respectively. For evaluation, we conduct comprehensive analytical and empirical analysis on a wide range of large models under both image and text data settings. These include (i) "Cross-scoring experiments", designed to elucidate how humans and LLMs evaluate each other's responses; (ii) "Exam scenario simulation", aimed at discerning the preferences of humans and LLMs in evaluating and filtering information; (ii) "AI-washing", demonstrating how generative models analyze, modify, and transmit information in a cyclical process. For experimentation, we curated specialized datasets comprising text and images, ensuring a rigorous performance evaluation.

Our **findings** highlight:

1. Large models tend to overrate their own answers while under-valuing human responses, which clearly indicates that synthesized data is more likely to prevail in information filtration processes.

2. For each cycle of information exchange between humans and large models, these models exhibit distinct preferences in amplifying or suppressing certain features. This behavior not only hinders performance enhancements but also complicates human intervention in the model's generative processes and information transmission.

3. It is worth noting that without ensuring a consistent presence of real human-generated data, large models may increasingly rely on self-generated datasets. This results in stagnating model performance. We term this phenomenon as the large model converging to a "local optimum", as elucidated in Section 5.2

The rest of the paper is organized as follows. In Section 2, we provide related work. In Section 3, we introduce notations and offer a brief background. In Section 4, we introduce Model Autophagy Analysis, for large models' self-consumption explanation. This is succeeded by comprehensive experimental evaluation and analyses in Section 5. Finally, we conclude our findings in Section 6.

## 2   Related Work

**Generative AI for information production and dissemination.** With widespread generative models, such as ChatGPT and DALL-E 3 Devlin et al. (2019); Radford et al. (2021); Betker et al. (2023) etc., anyone can interact with AI using natural language to express their feelings and/or posit different requirements. The AI uses multiple models to understand the content followed by utilizing various resources to generate and curate information (Kaddour et al.,

2023; Yin et al., 2023), which is later disseminated through the internet. This break-through has significantly altered the role of AI in human society. Generative models are no longer just simple tools, rather they have become a crucial component in the production and dissemination of information (Goldstein et al., 2023).

We emphasize that the risks associated with generative AI are not solely due to the biases and hallucination Huang et al. (2023b); Shen et al. (2023), they also stem from how humans interact with these systems, and the potential consequences, e.g., the creation of "information cocoons" Piao et al. (2023).

**Self-Training and Self-Consuming Models.** Recently, there has been a surge in use of automated routines/models for model alignment, data filtering, and data enhancements Gulcehre et al. (2023); Li et al. (2023), helpful to avoid the significant costs associated with creating humanly annotated data sets. A large number of LLMs-generated datasets are being used to fine-tune pre-trained foundation models (Taori et al., 2023; Xue et al., 2023). Simultaneously, the most powerful models currently available are often used as judges in numerous competitions (Bai et al., 2022a; Chiang et al., 2023). The risk involved in these approaches is significant, as models have already been preliminarily proven to be biased rather than impartial (Buolamwini & Gebru, 2018a; Wu & Aji, 2023; Liu et al., 2023). Alemohammad et al. (2023) proposed an autophagous loop for the computer vision models. Their work, characterized by the models trained using data generated by the models themselves, led to a decline in model performance and data diversity (Shumailov et al., 2023a). Subsequent studies have demonstrated similar traits in language models (Briesch et al., 2023).

## 3 Background/Preliminaries

**Notations:** In this paper, we use $d$ to represent the domain of QA-pair, helpful for providing question-specific context; $Q$ to represent the question; $D$ to represent doc-level information helpful for answering $Q$; $A$ to represent the human response/answer; $A_{m_i}$ to represent the response generated by model $m_i$, with $i \in \{1, 6\}$ represent six different LLMs; $A_{m_i s_5}$ to represent the highest quality response by model $m_i$; and $A_{m_i s_1}$ to represent the lowest quality response by model $m_i$.

### 3.1 Autophagous Loops

We redefine the relationship between large models and human societal information dissemination by drawing inspiration from the classic communication theory of the Ritual view by Carey (2008) (see Appendix A for details). This is illustrated in Figures 1 and 2, where we show both large models and humans can act as generators and filters of information in the Human-AI communication system. However, this system is prompting machine learning algorithms to encode all the stereotypes, inequalities, and power asymmetries that exist in human society (Birhane, 2022). For instance, women with darker skin are more likely to be misclassified in gender classification compared to men with lighter skin, which is due to the majority of samples in the training datasets being subjects with lighter skin tones (Buolamwini & Gebru, 2018b). The biased information generation and transmission process of large models and humans will further exacerbate these phenomena.

## 4 Model Autophagy Analysis

In this work, we propose Model Autophagy Analysis, for large models' self-consumption explanation. The core contributions of the model include: (1) designing realistic models of autophagy by drawing conclusions from human behaviors in utilizing large models; (2) curation of novel datasets (both text and image) followed by rigorous experimental studies to demonstrate/showcase how real-world data distribution is influenced by large models.

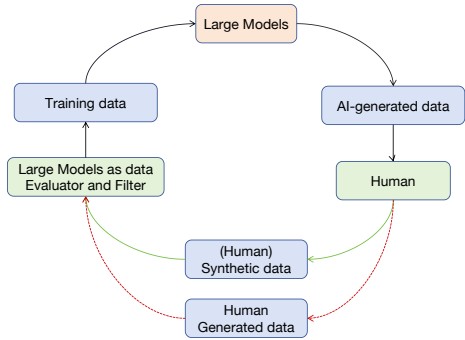

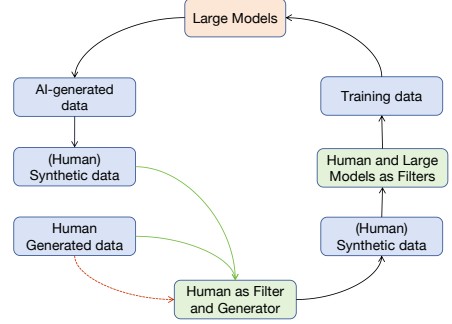

Figure 1: Self-consumption loop of large models. This figure is based on recent workflows for automated data generation and filtering Wang et al. (2023); Li et al. (2023). We emphasize the preferential nature of large models as generators and filters of synthetic data.

Figure 2: Self-consumption loop emphasizes the role of humans as filters and transmitters of information Veselovsky et al. (2023) while interacting with large models. Such a role primarily exists during the process of information dissemination in human society.

## 4.1 Self-consumption Loops

The proposed model uses two different models for *"self-consumption"* analysis (relying on autophagous loops) to simulate the interaction between humans and large models. We argue unlike previous work (Alemohammad et al., 2023), our self-consumption loops offer a more realistic and natural setting.

The end goal of our work is to understand different biases incurred by using humans and/or LLMs as generators and transmitters of information. These biases could help us understand the loss in data quality and/or diversity for LLM-generated and human-societal datasets. It also explicates the role these models play in the exchange of information within human society from a broader perspective. The self-consumption loops of large models and humans are explained as follows.

**Large Models.** Figure 1 outlines the cyclical influence of large models in the data processing life-cycle. As demonstrated in the Figure, the training data undergoes a transformation through either algorithmic refinement and/or human curation, resulting in what we term "synthetic data", see the "AI-generated data - Human - Synthetic data pathway" (shown as input to large models via solid green color). This contrasts with the "human-generated data", which primarily originates directly from humans and is typically less structured (shown as input to large models via a dotted red line).

We claim that large models that have access to both types of datasets are usually biased to filter preferentially, and/or prefer synthetic data over human-generated data for future learning cycles. This bias may either be inherently preferential or is incurred by the model training objective. To empirically validate this claim, we report multiple experiments in Section 5.

Furthermore, Figure 1 also explicates that the human's role in this cycle is not entirely passive. Humans, influenced by the outputs of large models, may unknowingly prioritize synthetic data due to its processed nature, which seems more immediately usable or relevant (Veselovsky et al., 2023). This preferential feedback loop can inadvertently lead to the diminishing the raw and/or human-generated data in the pool of data resources for being perceived as less refined.

**Humans.** Figure 2 showcases the specific behaviors of humans when interacting with large models. It presents a more detailed view for human-agent interactions, highlighting the fact that humans tend to favor the data generated by large models. This is illustrated in the Figure by using solid green lines showing high preference compared to the red dotted line, indicating inhibition. This fact/claim is also validated by our later rigorous experimentation,

which emphasizes that without transparent data provenance, humans may prefer outputs of large models, thus contributing to the cyclical bias toward synthetic data.

Correlating these two figures, we infer that the relationship between these two loops is symbiotic, offering a microcosm of humans' preferences in the Human-AI communication.

### 4.2 Rationality and Risks

In order to understand the rationale of the self-consumption loops on the data quality vs. diversity and examine the risks imposed, our study performs comprehensive experimentation. Note that a core proposition of our proposed models is that large models and humans cannot maintain objectivity and impartiality as part of the information dissemination loop, which could also be indicated by the colored line segments in Figures 1 and 2. In order to validate this proposition, we performs the following different analyses:

**Cross-Scoring Experiment.** Cross-scoring experiments aim to demonstrate the inhibitory and promotive phenomenons of information transmission within autophagous loops. We focus on whether LLMs and humans remain impartial while filtering and transmitting information and, if not, what kind of bias they induce. For this, we employ mainstream LLMs to generate question-answer pairs based on prompts (further details can be found in Appendix B.1), and instructed them to perform cross-scoring, i.e., using one model to evaluate and assign scores to the response generated by other models, etc.

In order to mitigate the impact of specified scoring standards, that is, tenths, percentages, and specific rules explained in Appendix B.3, we designed a simulated testing scenario to analyze which is more likely to prevail in the cycle of information dissemination in real-world scenarios: human-generated or AI-generated answers. This analysis aims to examine the consistency and the bias of humans and language models in adhering to these scoring standards.

**Exam Scenario Simulation.** Exam scenario simulation aims to answer the following question: Human-generated or AI-generated answers, which one wins in information screening and filtering? For this, we use LLMs and humans to simultaneously act as the generators and evaluators, see Section 7 for process-flow.

In this simulation scenario, the answers generated by LLMs alongside those produced by humans, are anonymized to mitigate any bias. To further eliminate the potential influence of the sequence in which the answers are presented, we also randomized their order. The language models and human participants, assuming the role of experts, were then asked to assign scores to these answers on a percentile scale and choose the best answer.

**AI-washing.** Finally, we conducted an "AI-washing" experiment in order to explore the risks posed by large models and humans as information generators, and to observe the changes in real data after multiple rounds of AI refinements. For these experiments, our goal is to analyze the trade-off between the information quality vs. diversity and comprehend large models' ability to enhance and weaken different information contents, i.e., whether large models are faithful messengers of information?

Basically, our aim is to answer the following questions: (i) Do these models effectively capture and convey the key information across different domains, such as identifying central themes in text and salient features in images? (ii) Do large models play the role of a link in the information transmission that may also lead to losses?

## 5 Experimentation

In this section, we perform comprehensive experiments to evaluate the preferences of humans and language models in information selection. Experimental details are as follows.

### 5.1 Experimental Settings

**Datasets.** For experimental evaluation, we curated three different datasets, namely: (i) QA-pairs, (ii) Book3, (iii) Image-ax. QA-pairs is a structured text data set used for cross-scoring and exam scenario simulation. Book3 and Image-ax are unstructured datasets used

for AI-washing experiments for text and images, respectively. Detailed description and statistics of the dataset is provided in Appendix C.1.

**Large Models.** We employed six different LLMs with varying architectures. These include ChatGPT (Li et al., 2022), GPT 4 (OpenAI, 2023), Claude 2[1], Llama-2-70b-chat (Touvron et al., 2023), PaLM2-chat-bison[2], and Solar-0-70b-16bit[3]. We opted for relatively large-scale models owing to their superior capabilities in instruction adherence, which we found lacking in small-scale models. In assessing textual diversity, we used the models bge-large-zh-v1.5 and bge-large-en-v1.5 by Xiao et al. (2023) as the embedding models. For computer vision tasks, we utilized the open-source model StableDiffusionXL (SDXL) by Podell et al. (2023).

**Evaluation Metrics.** For cross-scoring and exam simulation experiments, we alternatively use humans and LLMs as response generators and/or evaluators. The corresponding template for using LLMs as evaluators is given in Appendix B.3. This scoring criteria aims to verify the objective nature of LLMs as evaluators, i.e., whether a model assigns a higher scores to their response or response generated by other models. For AI-washing experiments, we measure the diversity in information after multiple rounds of iterations. Specifically, for the text data, we use cosine similarity (shown in Appendix C.2) to measure the diversity in the original text and the text reproduced by the LLM. For the image dataset, we visually analyze which features are preserved, omitted, and transformed by the LLMs.

**Experimental Setup.** For AI-washing experiments, the number of iteration rounds $N$ for text and images is 20. However, we find that for the text, the large model will no longer make significant changes to the text when $N > 4$. For images, however, the large model performs significantly differently across samples, as we discuss in detail later.

## 5.2 Experimental Results and Analysis

### 5.2.1 Cross-scoring Experiment.

The results of the cross-scoring experiment for QA-pairs are shown in Table 1. For these results, we report the scores assigned by LLMs and human annotators under the cross-scoring setting. For scoring via LLMs, we prompt each model to assess not only the answers generated by other models but also those produced by humans. The scoring range is between 1 to 5 (see Appendix B.3), with 1 indicating low quality and 5 for best quality answer. For human evaluation, we employed fifty crowd-sourced annotators to rate all question-answer pairs based on the scoring criteria in Appendix B.4. Note, we recorded the average scores for all samples, excluding instances where the LLMs and/or human evaluators refused to respond. These results indicate:

**Bias in Information Selection.** These results help us understand that LLMs do have an inherent capability to comprehend specific criteria of the prompt in Appendix B.3 and can adjust the quality of their generated answers based on relevant instructions.

However, we observe that each model exhibits certain preferences. Specifically, models tend to assign higher scores to the high-quality answers generated by themselves, particularly for ChatGPT and GPT 4, which both demonstrate high confidence in their own outputs. Also, we find that ChatGPT and GPT 4 exhibit similar characteristics in scoring; they are inclined to extreme scores, i.e., high or low scores (assigning scores of 1 or 5 with very high probability), at the same time. They both tend to assign lower scores to Claude 2 and PaLM2-chat-bison. These results are aligned with earlier research by Liu et al. (2023) that emphasized that current top-performing LLMs (both black-box and white-box) are narcissistic evaluators.

While evaluating the abilities of LLMs as answer/response generators, ChatGPT's worst-quality answers, which are easy to figure out for humans via crowd-sourcing, can still deceptively obtain higher scores from other models. Furthermore, we observed that Claude 2 tends to favor neutral and less-controversial ratings, often assigning average scores, i.e., 3 or 4, even for the worst quality answers. This is also reflected by a relatively higher

---

[1] https://www.anthropic.com/index/claude-2
[2] https://blog.google/technology/ai/google-palm-2-ai-large-language-model/
[3] https://huggingface.co/upstage/SOLAR-0-70b-16bit

| Scorer / Generator | ChatGPT | GPT4 | Claude 2 | Llama-2-70b-chat | PaLM2-chat-bison | Solar-0-70b-16bit | Human | Average |
|---|---|---|---|---|---|---|---|---|
| | | | | ORIGINALLY GENERATED ANSWER | | | | |
| ChatGPT | 4.33 | 4.29 | 3.88 | 4.25 | 3.92 | 4.17 | 2.48 | 3.90 |
| GPT4 | 4.63 | 4.56 | 4.04 | 4.41 | 3.95 | 4.60 | 2.77 | 4.14 |
| Claude 2 | 3.92 | 3.97 | 4.00 | 4.00 | 3.95 | 3.97 | 3.36 | 3.88 |
| Llama-2-70b-chat | 3.91 | 3.99 | 3.82 | 4.00 | 3.61 | 3.90 | 3.23 | 3.78 |
| PaLM2-chat-bison | 3.99 | 4.05 | 3.72 | 4.22 | 3.60 | 3.77 | 3.57 | 3.85 |
| Solar-0-70b-16bit | 4.10 | 4.35 | 4.05 | 4.16 | 4.01 | 4.12 | 2.59 | 3.91 |
| Human | 4.75 | 4.79 | 4.50 | 4.18 | 4.28 | 4.17 | 3.58 | 4.32 |
| | | | | BEST QUALITY ANSWER | | | | |
| ChatGPT | 4.24 | 4.28 | 4.41 | 3.80 | 4.21 | 4.20 | - | 4.19 |
| GPT4 | 4.52 | 4.75 | 4.20 | 4.11 | 4.00 | 4.36 | - | 4.32 |
| Claude 2 | 3.92 | 3.98 | 4.21 | 4.20 | 4.01 | 3.97 | - | 4.04 |
| Llama-2-70b-chat | 3.91 | 4.03 | 4.26 | 4.07 | 4.30 | 3.95 | - | 4.09 |
| PaLM2-chat-bison | 3.98 | 4.23 | 4.42 | 3.84 | 4.26 | 3.98 | - | 4.12 |
| Solar-0-70b-16bit | 4.34 | 4.43 | 4.42 | 4.33 | 4.28 | 4.11 | - | 4.32 |
| Human | 4.23 | 4.92 | 4.30 | 4.20 | 4.07 | 4.26 | - | 4.33 |
| | | | | WORST QUALITY ANSWER | | | | |
| ChatGPT | 3.13 | 1.33 | 1.27 | 1.27 | 2.83 | 2.21 | - | 2.01 |
| GPT4 | 3.19 | 1.40 | 1.29 | 1.33 | 2.98 | 1.70 | - | 1.98 |
| Claude 2 | 4.08 | 3.23 | 3.71 | 1.76 | 3.85 | 3.77 | - | 3.40 |
| Llama-2-70b-chat | 2.69 | 1.06 | 2.17 | 1.78 | 2.27 | 2.11 | - | 2.01 |
| PaLM2-chat-bison | 2.65 | 1.23 | 1.28 | 1.69 | 2.73 | 2.31 | - | 1.98 |
| Solar-0-70b-16bit | 3.28 | 1.26 | 1.89 | 2.40 | 2.37 | 2.40 | - | 2.27 |
| Human | 1.76 | 2.31 | 1.24 | 1.33 | 2.00 | 1.82 | - | 2.09 |

Table 1: Cross-scoring experiment results of language models and humans. The results in this table are average scores out of a five-point scale, assigned by both models and human evaluators, to the generated answers. These scores are calculated based on the criteria outlined in Appendix B.3, and Appendix B.4. The table is organized into three sections: with (i) "Originally Generated Answer", representing the scores for original response; (ii) "Best Quality Answer", representing scores ($A_{m_i s_5}$); and (iii) "Worst Quality Answer", representing scores ($A_{m_i s_1}$). We highlight the highest scores in each row in green and the lowest scores in red.

score assigned by Claude 2 to low-quality answers compared to other models. Also, the difference between its score for initial answers and best-quality answers is not significant. Llama-2-70b-chat and Solar-0-70B-16bit exhibit similar scoring and generation behaviors, which may be attributable to the fact that Solar-0-70B-16bit is fine-tuned form Llama-2-70b-chat with only Orca-style (Mukherjee et al., 2023; Lian et al., 2023) and Alpaca-style dataset (Taori et al., 2023), indicating that the pre-trained model has a significant influence on the model's preferences.

We also observed that prompting the model to generate better answers does not always lead to higher scores in our experiments. Only Claude 2 showed significant improvement in the answers compared to the original responses. Conversely, when we prompt model to generate poorer answers, the quality of answers generated by almost all LLMs in Table 1 decreased significantly. However, ChatGPT and Palm2-chat-bision still achieved relatively high scores, a possible reason could be attributed to the models being highly aligned to avoid producing harmful outputs (Lambert et al., 2022), so irrespective of input prompts, they always try to positively answer the question. We leave further investigation as a future research direction.

Furthermore, the scoring behavior of crowd-sourced annotators toward the answers generated by the large language model was largely consistent. This may be due to the highly structured and standardized nature of answers produced by AI systems aligned with human feedback. In contrast, human-generated answers received lower scores from the human annotators, with significant variations among different evaluators. This may be attributed to the higher degree of alignment between the large model and the human collective compared to alignment among individual humans.

### 5.2.2 Exam Scenario Simulation.

The exam simulation scenario aims to mitigate the impact of scoring criteria on the evaluative capabilities of large models and human evaluators in the previous experiments. It helps to understand the preferences of LLMs and humans in evaluating and filtering information (see Appendix 7 for process-flow). For this experiment, we used LLMs (ChatGPT, Claude 2) and humans to simultaneously act as generators and evaluators. The results of the exam scenario simulation are shown in Table 2. We report the average scores assigned by

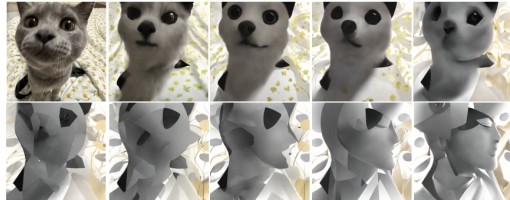

Figure 3: An example illustration of AI-washing on images that shows that repeatedly processing images $N$-times ($N$=1:5) using SDXL model (Podell et al., 2023) may lead to serious biases.

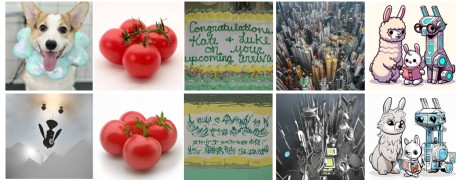

Figure 4: After 20 rounds of AI-washing experiments with the SDXL model (Podell et al., 2023), it becomes evident that different images retain and discard details in varying manners.

evaluators for the cases when their responses were selected as the best answers. In addition, detailed percentile scores for this experiment are provided in Appendix C.3. These results show:

**LLMs win in information screening.** Responses from LLMs consistently garnered acceptance from evaluators and were frequently chosen as the best answers. In contrast, human-generated responses were rarely chosen as the best response, indicating a challenge in integrating authentic human-generated responses into models' training data and the real-world human feedback loop.

This finding substantiates the potential risks also highlighted in Section 4.2 that emphasized that human-generated responses typically receive comparatively lower consideration in the self-consumption loops.

### 5.2.3 AI-washing.

Example demonstration of AI-washing experiments for text data set processed via ChatGPT are shown in Appendix D.1 (Table 5). The prompts used for these experiments are provided in Appendix B.2. Likewise, some examples of AI-washing experiments for images are shown in Figure 3 and 4. For this, we instruct SDXL (Podell et al., 2023) to process these samples $N$ times. These results show:

**LLMs are Biased Information Transmitters.** Large models are inherently biased regarding the manner and content of conveyed information. For instance, textual example (in Table 5) processed $N$ times ($N$=1:5) by ChatGPT shows subtle shifts in the sample's language style and narrative technique. Similarly, for images in Figure 3, the model preserved the color distribution of the original image but altered the main subject from a cat to a human portrait. Conversely, for the images in Figure 4, we observed a dominant alteration in image style rather than a change in the primary content for the first, fourth, and fifth images. Whereas, the images at the second and third positions underwent several iterations with minimal transformation, with the third image lost readability for the textual content.

This inconsistency may be attributed to the fact that while SDXL is renowned for generating high-quality images, the definition of quality in the context of generative models is subjective and heavily influenced by the annotations of the training dataset, also explained in detail

| Generator | Evaluator | | |
|---|---|---|---|
| | ChatGPT | Claude 2 | Human |
| AVERAGE SCORE | | | |
| ChatGPT | **95** | **90** | **91.7** |
| Claude 2 | 92 | 88 | 90 |
| Human | 90 | 75 | 80 |
| SELECTED AS BEST ANSWER | | | |
| ChatGPT | 41 | **58** | **66** |
| Claude 2 | **55** | 42 | 25 |
| Human | 4 | 0 | 9 |

Table 2: Result for exam scenario simulation. We boldface the best scores.

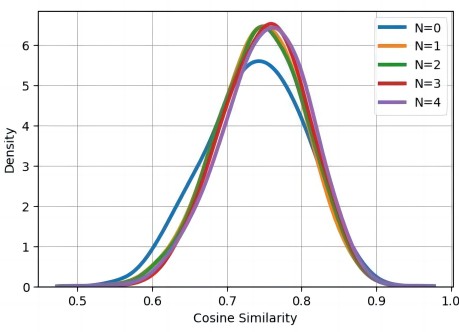 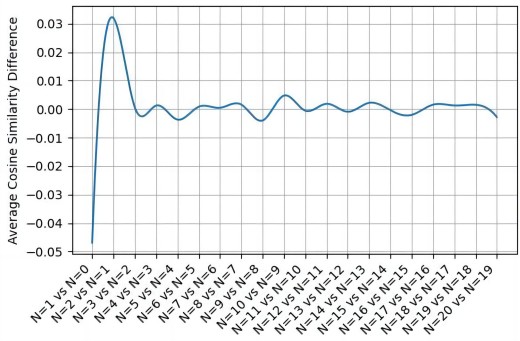

Figure 5: Density distributions of cosine similarity scores for text samples from Book3 processed $N$ times by ChatGPT.

Figure 6: We report difference in cosine similarity for $N$ successive iterations ($N \leq 20$). For this graph, we use samples from Book3 dataset and ChatGPT as LLM.

by Podell et al. (2023). These observations indicate a bias in the model's processing, where certain features are selectively preserved or altered based on the model's training objective and inherent design. This results in a higher probability of generating images containing specific features and/or information, e.g., loop, such as hand-drawn styles, portraits, close-ups of objects with clear backgrounds, etc.

To summarize, these findings demonstrate that large models are inherently biased regarding the manner and content of conveyed information. And repeated processing of images with generative models is akin to information and feature filtration, where generative models tend to emphasize or de-emphasize certain features.

**Information Diversity.** In order to further understand the impact of model training on data quality and diversity, we analyze the training behavior of large models. Specifically, we analyze the cosine similarity scores for multiple iterations of humanly-generated and LLM-generated text. Corresponding results in Figure 5 shows that for successive rounds of text processed using LLMs, the cosine similarity between texts increases significantly, with the lower tail (i.e., line ($N = 0$) with cosine similarity b/w 0.5 and 0.7) being washed out. This shows that while acting as information disseminators, the large models exhibit some unique characteristics, i.e., they tend to optimize more for real-world data while being more lenient towards self-generated samples.

We also analyzed the difference in cosine similarity across successive runs. The line graph in Figure 6 shows that after approximately three iterations, the difference in the average cosine similarity across multiple rounds tends to stabilize. With the growing utility of large models, especially as a mechanism for knowledge/information comprehension, this phenomenon poses significant risks to the fairness and diversity of information dissemination.

**Local Optimum.** To re-emphasize the results in Figure 6, we observe that starting from the third iteration, the samples appear to have reached a "local optimum", requiring fewer updates in model parameters. This highlights that: (i) By the third iteration, the large model has significantly adapted to the data details and/or style; (ii) While the LLM effectively extends its knowledge about the original content, it somehow falls short of achieving self-styling changes and updates. This self-bias suggests that when processing information, the large models learn to rewrite the text of different styles into a uniform style rather than enhancing the quality and diversity of the text also mentioned by Xu et al. (2024). This tendency could lead the model to become entrenched in its own style, potentially limiting its adaptability and creativity.

# 6 Conclusion

In this work, we propose two different self-consumption loops to examine large models as generators and disseminators of information within human society. Results emphasize

that AI-generated information tends to prevail in information filtering, whereas real human data is often suppressed, leading to a loss of information diversity. This trend limits next-generation model performance owing to fresh data scarcity and threatens the human information ecosystem. Some of the limitations of our current work are mentioned in Appendix E.

**Acknowledgements.** Di Wang, Lijie Hu and Muhammad Asif Ali are supported in part by the baseline funding BAS/1/1689-01-01, funding from the CRG grand URF/1/4663-01-01, FCC/1/1976-49-01 from CBRC and funding from the AI Initiative REI/1/4811-10-01 of King Abdullah University of Science and Technology (KAUST). Di Wang is also supported by the funding of the SDAIA-KAUST Center of Excellence in Data Science and Artificial Intelligence (SDAIA-KAUST AI). We also acknowledge support from OpenAI API Researcher Access Program.

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

# A    Ritual View of Communication

Carey (2008) conceptualized *"The Ritual View of Communication"* in his communications theory. They emphasized that communication is not just a medium for the transmission of information, but as a symbolic process that contributes to the construction and maintenance of social reality.

Carey's theory posits that communication is integral to the representation, maintenance, adaptation, and sharing of a society's cultures over time. In short, the Ritual view conceives communication as a process that enables and enacts societal transformation. This theory even extends to modern media forms such as newspapers and social media platforms in our modern communication age (Thornburg, 1995; Edwards et al., 2016). The emergence of the internet and social platforms (e.g., Facebook, Twitter etc.,) has further developed the ritualistic nature of communication. These advancements have facilitated the growth of global online communities by redefining their patterns of interaction (Jain et al., 2021; Lee & Kim, 2014).

Similarly, generative-AI represents a profound transformation in the modes of human social communication and the ways humans interact with AI (Fui-Hoon Nah et al., 2023; Rane, 2023). We should regard AI, trained on extensive human civilization data, as an integral part of human societal information transmission, acknowledging its role in shaping and sharing the cultural and social implications of human society (Papadimitriou, 2016; Rane, 2023).

## B  Prompt Templates

### B.1  Answer Generation Prompt Template

In this section, we present the prompt template for generating the `Originally Generated Answer`, the `Best Quality Answer` and the `Worst Quality Answer`.

---

**The prompt template for generating the `Originally Generated Answer`**

Question:query+detail
Answer the question:

---

**The prompt template for generating the `Best Quality Answer`**

Below is an instruction from an user and a candidate answer. Evaluate whether or not the answer is a good example of how Al Assistant should respond to the users instruction. score=5: It means it is a perfect answer from an Al Assistant. It has a clear focus on, being a helpful Al Assistant, where the response looks like intentionally written to address the user's question or instruction without any irrelevant sentences. The answer provides high-quality content, demonstrating extensive knowledge in the area, is very well written, logical, easy to follow.

Question: query+detail

Now give an example of an Al assistant answer with a score of 5 about the question:

---

**The prompt template for generating the `Worst Quality Answer`**

Provide an AI assistant response with a score of 1(lowest quality) based on the given instruction: Your example should demonstrate an incomplete, vague, off-topic, controversial, or exactly what the user asked for.

Question: query+detail

Now give the counter-example of an AI

Assistant response:

---

### B.2  AI-Washing Prompt Template

We use prompts that have nothing to do with the content generated and instead have to do with the quality of the generation, as presented in the following table.

---

**The prompt template for AI-washing**

**Prompt for ChatGPT:**
(en) Polish the following paragraph:
{paragraph}

**Prompt for SDXL:**
Positive: best quality, masterpiece, ultra detailed, 8K, UHD, Ultra Detailed
Negative: worst quality, split picture, ignoring prompts, lowres

---

## B.3 LLM Cross-scoring Prompt

We use the same prompt as in the work of Li et al.. as shown in the following.

---

**The prompt template for evaluating answers**

Below is an instruction from an user and a candidate answer. Evaluate whether or not the answer is a good example of how Al Assistant should respond to the users instruction

Please assign a score using the following 5-point scales

1: It means the answer is incomplete, vague, off-topic, controversial, or exactly what the user asked for For example, some content seems missing, the numberedlist dnot start from the beginning, the opening sentence repeats the user's question. Or the response is from another person's perspective with their personal experience (e.g. taken fmblog posts), or looks like an answer from a forum. Or it contains promotional text, navigation text, or other irrelevant information

2: It means the answer addresses most of the asks from the user. It does not directly address the user's question. For example, it only provides a high-level instead of the exact solution to the user's question

3: It means the answer is helpful but not written by an Assistant. It addresses the basic asks of the user. It is complete and self-contained with the drawback that the response is not written from an assistant's perspective, but from other people's perspective. The content looks like an excerpt from a blog post, or web page, and provides search results. For example, it contains personal experience or opinion, mentions comments section, or shares on socialmedia, etc.

4: It means the answer is written from an Al assistant's perspective with a clear focus on addressing the instruction. It provides a the complete, clear, and comprehensive response to user's question or instruction without missing or irrelevant information. It is well organized self-contained, and written in a helpful tone. It has minor room for improvement, more concise and focused.

5: It means it is a perfect answer from an Al Assistant. It has a clear focus on, being a helpful Al Assistant, where the response looks like intentionally written to address the user's question or instruction without any irrelevant sentences. The answer provides high-quality content, demonstrating extensive knowledge in the area, is very well written, logical, easy to follow, engaginIt means it is a perfect answer from an Al Assistant. It has a clear focus on, being a helpful Al Assistant, where the response looks like intentionally written to address the user's question or instruction without any irrelevant sentences. The answer provides high-quality content, demonstrating extensive knowledge in the area, is very well written, logical, easy to follow, engaging, and insightful please first provide brief reasoning you used to derive the rating score, and then write "Score: —rating" in the last line.

generated instruction: {question}+{detail}

answer: {answer}

---

## B.4 Scoring Criteria for Human Evaluation

Based on the modifications to the previous scoring prompts for the LLMs, we created scoring criteria for our crowdsourced annotators, as demonstrated in the following.

---

**Scoring criteria for crowd-sourced annotators**

You are to evaluate the quality of a response given to a specific question. Your evaluation should consider how well the response addresses the query, its completeness, clarity, and relevance.

Scoring Scale:

Score 1: The response is unsatisfactory. It is incomplete, vague, unrelated to the question, or may simply echo the question without providing an answer. The content may be off-topic, contain promotional material, or resemble a personal opinion rather than a factual answer.

Score 2: The response generally relates to the question but does not directly answer it. It may provide an overview rather than the specific details or solution that the question warrants.

Score 3: The response is useful and addresses the basic query. However, it may not be from the expected perspective, potentially reading like a generic excerpt from a blog or an article rather than a targeted answer.

Score 4: The response is on target, addressing the question directly and completely with a clear and organized presentation. Minor improvements could be made to enhance focus or conciseness.

Score 5: The response is exemplary, directly and comprehensively addressing the question with high-quality content. It demonstrates extensive knowledge, is logically structured, easy to understand, engaging, and provides insight. Procedure for Evaluation:

Read the question and the corresponding response carefully. Evaluate the response based on the above criteria.

Question: query+detail

Response: answer

Record your score :

---

## C   Experimental Details

### C.1   Dataset

For evaluation, we manually curated text and image data sets. Details about these datasets are provided as follows.

**QA-pairs.** For this, we initially handpicked 100 diverse question-answer pairs from Stack-Overflow and Quora as the seeds. Subsequently, for each question in these pairs, we used large models to generate initial responses with instruction in Section B.1. We manually screened the most answered questions in Stack-Overflow and Quora, including psychology, books, mathematics, physics, and other fields. At the same time, we selected fragments from the novel corpus for anonymization processing to study the behavior of the language model when delivering real human-generated data.

These responses were further processed to curate datasets rated from 1 (lowest) or 5 (highest) in terms of quality, similar to the self-alignment approach proposed by  Li et al. (2023). The prompts used for generating these diverse responses are detailed in Appendix B.1. Final data set encompasses approximately 1,900 question-answer pairs. Table 3 illustrates the

distribution of the data. Formally, the dataset consists of a series of 22 tuples, each structured as follows:

$$T_j = \{d, Q, D, A\} \cup \bigcup_{i=0}^{5} \{A_{\mathrm{m}_i}, A_{\mathrm{m}_i \mathrm{s}_5}, A_{\mathrm{m}_i \mathrm{s}_1}\} \tag{1}$$

| Data Category | Percentage |
|---|---|
| Stackoverflow QA | 30% |
| Quora QA - Books | 10% |
| Quora QA - Psychology | 10% |
| Quora QA - Life | 10% |
| Quora QA - Happiness | 10% |
| Quora QA - Personal Experiences | 10% |
| Quora QA - Mathematics | 10% |

Table 3: Categories for QA-pairs

**Book3.** For AI-washing experiment in Section 5.2.3, the raw text dataset construction process begins with the selection of passages from classic literature known for their rich stylistic features and thematic significance, where the English dataset is excerpted from the pile books3 (Gao et al., 2020), and the Chinese passages are selected from WebNovel. A meticulous anonymization process is employed to prevent the large language model from identifying the textual sources. This involves the alteration of recognizable names, places, and events.

**Image-ax.** The image dataset was constructed by selecting a subset of images from the ILSVRC data (Russakovsky et al., 2015) and web resources. We select categories covering a wide range of topics and scenarios, in order to cover a broad range of visual features and complexity. On the visual dataset, we sampled and cleaned the ILSVRC Russakovsky et al. (2015) to ensure the diversity of image clarity and classification.

C.2  Computational details

**The density of cosine similarity scores** between two vectors $A$ and $B$ is calculated as:

$$\text{Cosine Similarity} = \frac{A \cdot B}{\|A\|\|B\|} \tag{2}$$

where $A$ and $B$ are the embedding vectors of two paragraphs.

The density of cosine similarity scores is estimated using Kernel Density Estimation (KDE), which is given by:

$$\text{KDE}(x) = \frac{1}{n} \sum_{i=1}^{n} K_h(x - x_i) \tag{3}$$

where,

- $K_h$ is the kernel function with bandwidth $h$
- $x$ represents the value at which the density is estimated
- $x_i$ are the data points (cosine similarity scores in this case)
- $n$ is the number of data points.

The KDE process smoothens the discrete data points to create a continuous density curve, represented on the y-axis of Figure 5.

**The average cosine similarity difference** between two successive iteration is calculated as follows:

$$\Delta S = \bar{S}_i - \bar{S}_{i-1} \tag{4}$$

Where:

- $\Delta S$ is the average cosine similarity difference between the current text and the previous text.
- $\bar{S}_i$ is the average cosine similarity for current text.
- $\bar{S}_{i-1}$ is the average cosine similarity for previous text.
- For the first file comparison, $\bar{S}_{i-1}$ is assumed to be 1.

This calculation method provides a metric for assessing the change in similarity across sequential data sets, reflecting the evolution or consistency of the data characteristics.

## C.3 Exam Scenario Simulation

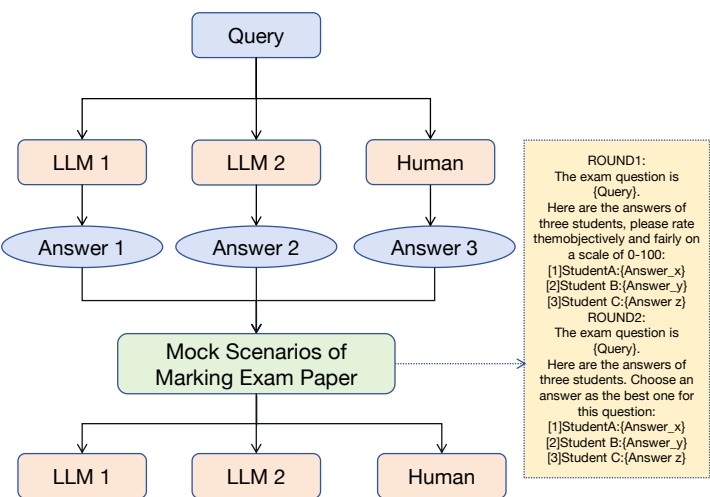

Figure 7: Exam Scenario Simulation

Figure 7 displays the flowchart and prompt template for the Exam Scenario Simulation experiment.

Table 4 compares the scores given by humans with the answers of other humans and to the models in different percentiles in this simulation of the exam scenario. The table illustrates the gap between the scores for humans and models widens as we move to higher percentiles. For example, at the 25th percentile, the gap is 22 points (85 vs. 63), while at the 100th percentile, the gap is 8 points (100 vs. 92). This indicates that human scoring of other humans' answers exhibits greater variability, meaning humans are more critical or less

| Percentile | Humans Scoring Humans | Humans Scoring Models |
|---|---|---|
| 25 th | 63 | 85 |
| 50 th | 75 | 88 |
| 75 th | 80 | 93 |
| 90 th | 83 | 96 |
| 100th | 92 | 100 |

Table 4: Comparison of Scores Given by Humans to Other Humans and to Models Across Different Percentiles.

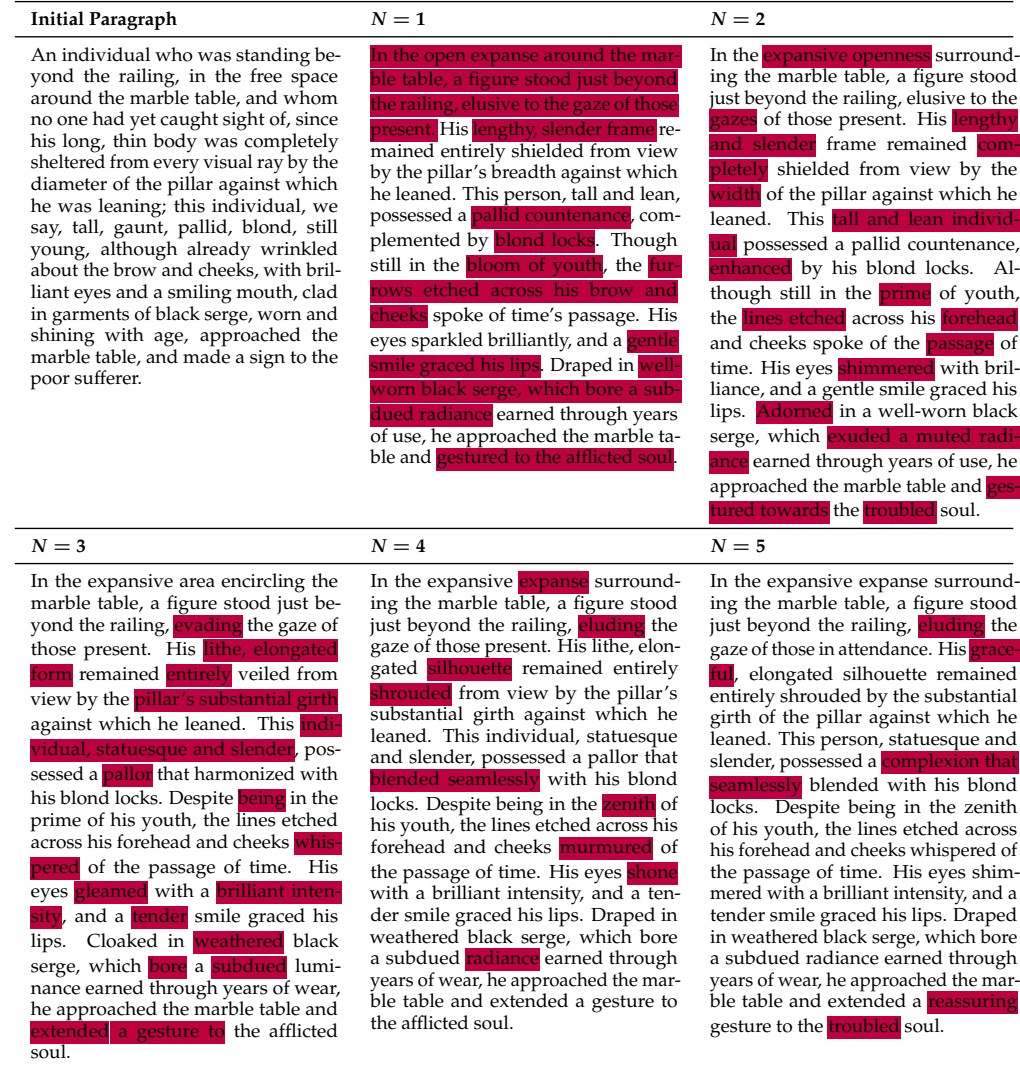

| Initial Paragraph | N = 1 | N = 2 |
|---|---|---|
| An individual who was standing beyond the railing, in the free space around the marble table, and whom no one had yet caught sight of, since his long, thin body was completely sheltered from every visual ray by the diameter of the pillar against which he was leaning; this individual, we say, tall, gaunt, pallid, blond, still young, although already wrinkled about the brow and cheeks, with brilliant eyes and a smiling mouth, clad in garments of black serge, worn and shining with age, approached the marble table, and made a sign to the poor sufferer. | In the open expanse around the marble table, a figure stood just beyond the railing, elusive to the gaze of those present. His lengthy, slender frame remained entirely shielded from view by the pillar's breadth against which he leaned. This person, tall and lean, possessed a pallid countenance, complemented by blond locks. Though still in the bloom of youth, the furrows etched across his brow and cheeks spoke of time's passage. His eyes sparkled brilliantly, and a gentle smile graced his lips. Draped in well-worn black serge, which bore a subdued radiance earned through years of use, he approached the marble table and gestured to the afflicted soul. | In the expansive openness surrounding the marble table, a figure stood just beyond the railing, elusive to the gazes of those present. His lengthy and slender frame remained completely shielded from view by the width of the pillar against which he leaned. This tall and lean individual possessed a pallid countenance, enhanced by his blond locks. Although still in the prime of youth, the lines etched across his forehead and cheeks spoke of the passage of time. His eyes shimmered with brilliance, and a gentle smile graced his lips. Adorned in a well-worn black serge, which exuded a muted radiance earned through years of use, he approached the marble table and gestured towards the troubled soul. |

| N = 3 | N = 4 | N = 5 |
|---|---|---|
| In the expansive area encircling the marble table, a figure stood just beyond the railing, evading the gaze of those present. His lithe, elongated form remained entirely veiled from view by the pillar's substantial girth against which he leaned. This individual, statuesque and slender, possessed a pallor that harmonized with his blond locks. Despite being in the prime of his youth, the lines etched across his forehead and cheeks whispered of the passage of time. His eyes gleamed with a brilliant intensity, and a tender smile graced his lips. Cloaked in weathered black serge, which bore a subdued luminance earned through years of wear, he approached the marble table and extended a gesture to the afflicted soul. | In the expansive expanse surrounding the marble table, a figure stood just beyond the railing, eluding the gaze of those present. His lithe, elongated silhouette remained entirely shrouded from view by the pillar's substantial girth against which he leaned. This individual, statuesque and slender, possessed a pallor that blended seamlessly with his blond locks. Despite being in the zenith of his youth, the lines etched across his forehead and cheeks murmured of the passage of time. His eyes shone with a brilliant intensity, and a tender smile graced his lips. Draped in weathered black serge, which bore a subdued radiance earned through years of wear, he approached the marble table and extended a gesture to the afflicted soul. | In the expansive expanse surrounding the marble table, a figure stood just beyond the railing, eluding the gaze of those in attendance. His graceful, elongated silhouette remained entirely shrouded by the substantial girth of the pillar against which he leaned. This person, statuesque and slender, possessed a complexion that seamlessly blended with his blond locks. Despite being in the zenith of his youth, the lines etched across his forehead and cheeks whispered of the passage of time. His eyes shimmered with a brilliant intensity, and a tender smile graced his lips. Draped in weathered black serge, which bore a subdued radiance earned through years of wear, he approached the marble table and extended a reassuring gesture to the troubled soul. |

Table 5: Example of AI-Washing experiment for text from ChatGPT. *N* represents the number of times the large language model is used for refinement, with each changed part highlighted.

consistent in agreeing with other humans' answers. In contrast, the human scoring of the model-generated answers is more consistent, suggesting a higher alignment between humans and the models than between individual humans.

# D   Examples of AI-washing Experiments

## D.1   AI-washing for text data

We report an example illustration for AI-washing for text data in Table 5.

## D.2   AI-washing for image data

We give more examples of the image AI-washing experiments in Figure 8 and Figure 9, where we can observe that after iterative processing the textual parts of the images are frequently changed and fragmented, e.g., the text on the airplane, the numbers on the clock, and the letters on the potato chip packet are changed several times. The pet dog is gradually stylized as a cartoon and becomes black and white, and the cauliflower is transformed by

the model into a bouquet of flowers after the first processing and is gradually stylized as a cartoon. At the same time, the model adds features to the initial image based on stereotypes from the training data, such as the logo of a clock and the logo of a car. In contrast, the overall structure, colors, and borders of the image of an apple are not significantly changed. It can be seen that the model will be affected by the model's own structure and training process when processing image features and has different enhancement or inhibition effects on different features.

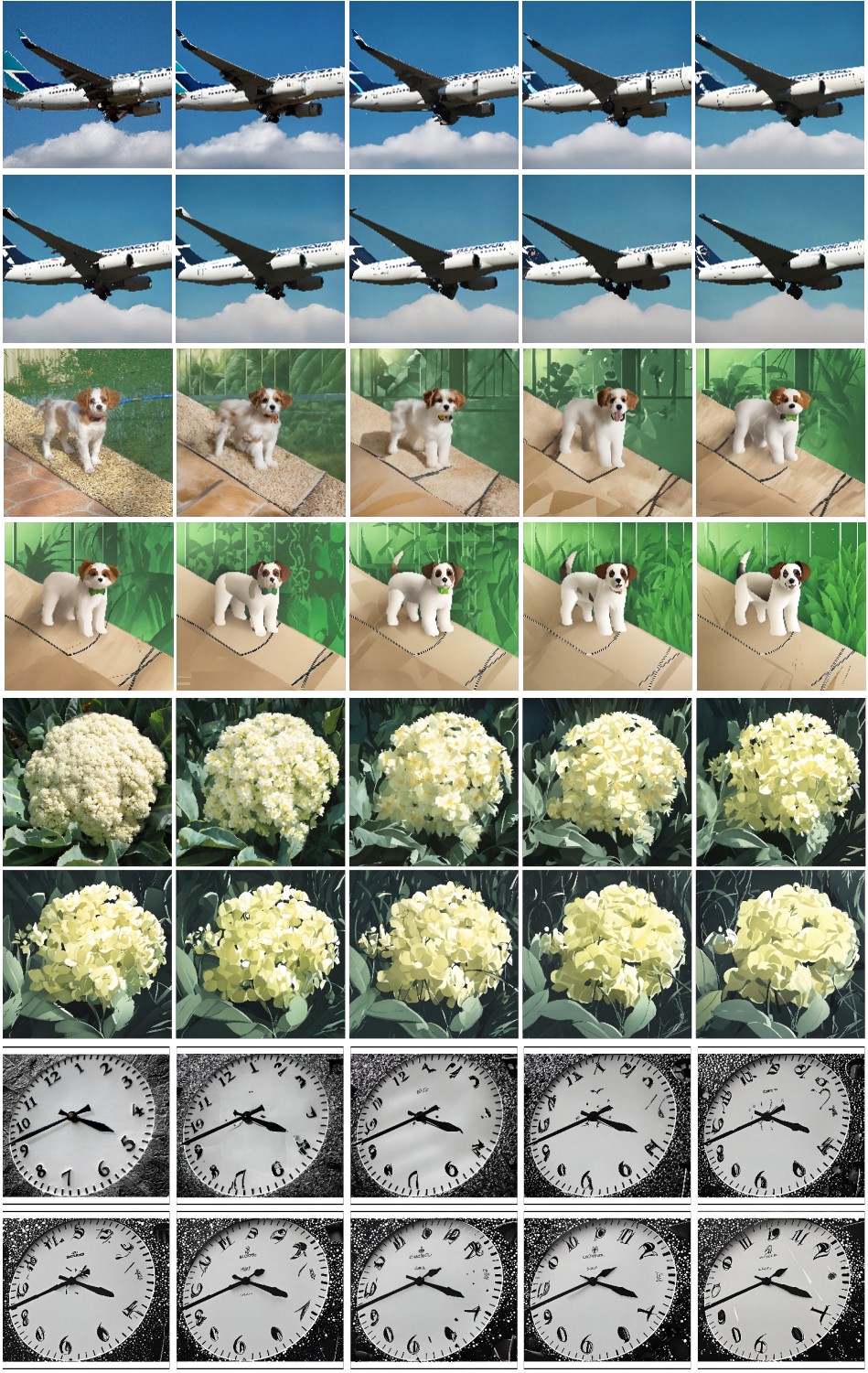

Figure 8: Examples of image AI-washing experiments (part1)

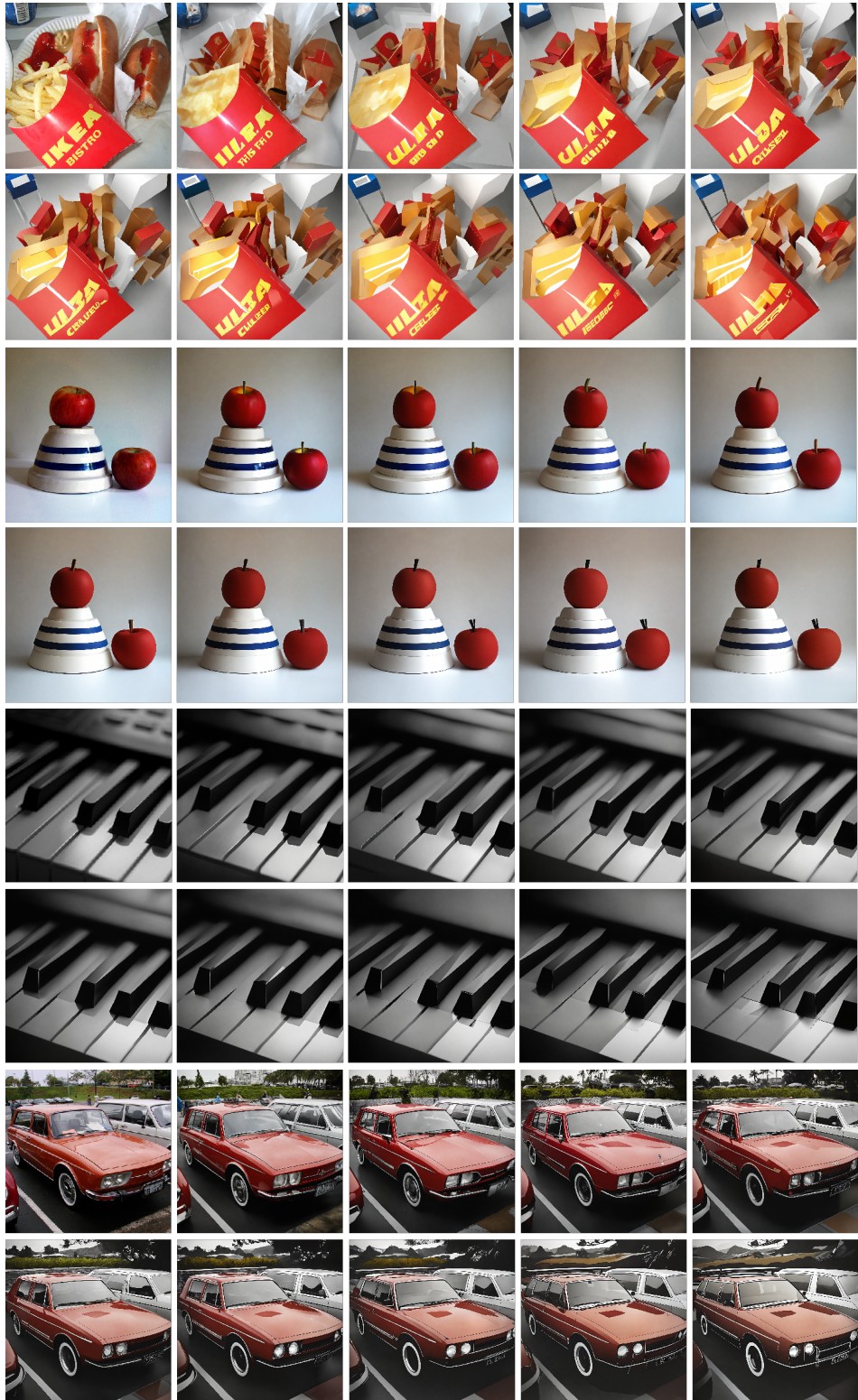

Figure 9: Examples of image AI-washing experiments (part2)

# E   Limitations

**Selection of Models.** While we have experimented with LLMs that are available, many outstanding models are worth exploring in the future. These include the GLM family of modelsDu et al. (2022), which are known for their innovative architectures, and the MoE-structured Mixtral 8x7B[4], etc. In addition, some open-source multilingual models are also worth investigating, such as the Qwen series of models trained on a large Chinese corpusBai et al. (2023) and the Arabic model Jais[5]. Models with different languages, parameter sizes, and architectures exhibit different behaviors. In the field of visual models, more open-source and commercial models, such as Midjourney and DALL-E 3, are worth investigating. In future research, we aim to deeply analyze the roles and characteristics of these models as an important part of human social information transfer.

**Reliability of crowd-sourced Annotators.** A significant portion of our conclusions is derived from crowd-sourced annotators sponsored by a start-up company's data annotation department. Of these annotators, 64 % hold graduate degrees in science and engineering, and all possess proficient bilingual reading skills in Chinese and English. However, ensuring that their existing AI knowledge does not bias their judgments remains challenging. Additionally, the distribution of our annotators in the real world varies from the general user base of generative models. There is also an ongoing debate about the reliability of crowd-sourced workers(Spurling et al., 2021; Tarasov et al., 2014). Veselovsky et al. (2023) have discussed the behavior of annotators using LLMs for labeling, which could compromise the reliability of the results.

---

[4]https://mistral.ai/news/mixtral-of-experts/
[5]https://inceptioniai.org/jais/

