# OpenReview forum: "Model Autophagy Analysis to Explicate Self-consumption within Human-AI Interactions"
_colmweb.org/COLM/2024/Conference — COLM_

### Official Review · Reviewer_vpdp · 2024-05-08

**Rating:** 7
**Confidence:** 4
**Ethics Flag:** 1

**Summary:**

The authors analyze the roles of humans and LLMs in producing and filtering data. They show that:
1) LLM-produced language tends to be preferentially-rated, by both models and humans, compared to human-produced language.
2) Repeated processing of inputs by AI models tends to distort the inputs and converge to a stable, but potentially local, minimum.

I found that the paper started out very well written, although the clarity of the writing seemed to worsen (to me) over the course of the paper.

I find this work potentially quite significant. The human and AI-rating results, in particular, seem extremely important for future AI research work.

The experiments are largely well done. While there are other experiments that I would like to see done, that does not detract from the current experiments.

**Questions To Authors:**

1. There are a few typos. E.g., a mis-capitalization at some point, "Claud2" instead of "Claude2", "Biasedness" instead of "Bias", mis-numbering of bullet points, etc.

2. Figure 6 could be improved by plotting the absolute value, which would then show a decreasing magnitude.

3. As noted in the "Reasons to Reject" section, I would like a bit more clarity on the role of information transmission vs. training on AI-generated data. Did the authors run experiments using AI-generated data?

4. The bolding in Table 1 confuses me. The caption states that "We bold-face the best scores in each section", but I see multiple bold-faced entries per (what I think of as a) section.

**Reasons To Accept:**

Overall, I find this work strongly motivated, important, and it provides empirical evidence of a subtle phenomenon. I think the experiments for human and AI-preference ratings to be very well done and compelling.
The authors have done a good job experimenting with multiple AI models, and for including lots of supporting details in appendices.

**Reasons To Reject:**

1) I think the greatest weakness of this work, in my mind, is clearly scoping the importance of information transmission by AI models. One half of the experimental results consider how AI models are biased information transmitters, as illustrated by drift over multiple iterations. My question is, why is this so important?

I think the authors are hinting at how 1) if AI models generate biased or skewed data and 2) such data are preferentially selected by humans or AI models for training then 3) training AI models on AI-generated data could result in worsening models stuck at local optima. The authors demonstrate 1 and 2 in their experiments, but I don't believe they run experiments with 3. Thus, claims about how training on AI-generated data should be carefully considered.

2) I found the framing of the paper slightly confusing. The authors introduce the acronym MONAL as if they are introducing a method; I believe this work could be more strongly positioned as simply a scientific finding about humans and AI. There's no need to try to reach beyond that and introduce a method.

---

> ### Author Rebuttal · Authors · 2024-05-31
>
> Thanks for your valuable feedback. Our response for each individual point is as follows:
>
> **Q1. Information transmission vs.Training on AI-generated data**
>
> * The impact of repeatedly training on AI-generated data has already been covered in literature, as mentioned in Section 2 of our work, stating that repeatedly training models on synthetic data can reduce data diversity and eliminate long-tail samples. The core focus of our work is to complement these studies by focusing more on the interaction between humans and large models, emphasizing the biases in information transmission and filtering during this interaction.
>
> **Q2. Monal Acronym**
>
> * In light of your feedback, we are considering renaming our paper to better reflect its content and purpose. We plan to update the name "Monal" to "a systematic framework for LLM-vs-human interaction analysis" to accurately convey the study's focus.
>
> **Q3. Typos:**
>
> * We have revised the typos, i.e., "Claud 2" to "Claude 2"; "biasedness“ in section 5.2.1 to "bias"; "iteration round" to "iteration rounds" in section 5.1; change "Chat-GPT" in Figure 6 to "ChatGPT".
>
> **Q4. Figure 6**
>
> * We will revise Figure 6 to plot absolute values, showing decreasing magnitude over time.
>
> **Q5. Information transmission vs.Training on AI-generated data**
>
> * For details, refer to our response to Q1.
>
> **Q6. Bold-face in Table 1**
>
> * We apologize for the confusion caused by our attempt to bold noteworthy data, which inadvertently decreased readability. In the final version, we will use a different color to highlight different aspects of the results and optimize the notation of Figure 1 for better clarity.
>
> **Reference**
>
> [1] Alemohammad, Sina, et al. "Self-consuming generative models go mad." ArXiv, 2023
>
> [2] Ilia Shumailov, Zakhar Shumaylov,et al. The curse of recursion: Training on generated data makes models forget. ArXiv,2023

---

> > ### Comment · Reviewer_vpdp · 2024-06-03
> >
> > Thanks for the reply. I accidentally replied to a different response. I will leave my rating unchanged.

---

### Official Review · Reviewer_tbTD · 2024-05-10

**Rating:** 7
**Confidence:** 4
**Ethics Flag:** 1

**Summary:**

This paper explores the use and downstream impact of data generated by large language models that might then be further fed back into them at later training stages. They term this as “model autophagy” and characterise the differences in the data generated by large models, how they evaluate/rate/have preferences over such generated data, how human-generated data/evaluators compare to this data, and the downstream impacts this could have in a world where model generated data is produced in large amounts and might impact the diversity of existing data on average. Their findings about preferences of models towards synthetic vs. human data are novel and extremely important to understand the effect of more and more model-generated data that is now publicly available on the internet and how this might affect future training of such models, as well as users who take in such input.

**Questions To Authors:**

1. The related works section looks great but the citations are a little misleading—they are often only referring to a new paper using the concept in question from 2023-24, not actually the original paper that introduced that concept. Unless there is a specific reason to cite that particular paper, it would make more sense to cite the paper that first introduced a concept, rather than just citing the most recent paper from the last year. E.g., (“judges in numerous competitions, Chian et. al., 2023 → this should also have Bai et. al., 2022; “chatgpt and dalle-2, Betker et. al., 2023 → this should have Radford et. al., Devlin et. al., etc as the first generative models)
2. In Section 4.2 in the cross-scoring experiment definition, can we clearly state how this is done across two model pairs, whether it is exhaustive over all pairs, and whether there is a human counterpart/experiment as a control?
3. Section 4.2 last paragraph, “do these models capture the main points in the information that needs to be conveyed?” → can we rephrase this to be more definitive e.g., specifying what it would be to “capture a main point” in the text vs. image domains and so on.
4. Section 5.2.1: in the results where models (ChatGPT and GPT-4) seem to prefer their own outputs, can we also provide intuition for this information theoretically/based on the models objective? E.g., for one of the models, if you scored perplexity under that model over something it generated vs. another human/models generations it should be lower, however this is separate from the autorater setting but might be linked to the fact the same model reframed as an autorater should theoretically prefer its own generation.
5. Results about how the scoring of human annotators towards AI systems is consistent have interesting implications that should be discussed more i.e., the fact that model generations result in text that is less diverse (than what would be produced by a population of humans) and seems to always veer towards some mean/normal and this is reflected in the human judgements towards them (roughly constant) vs. human judgments over other human generations (highly variable).
6. For Table 2, it would be really interesting to see a graph of the distribution of scores produced for humans scoring humans vs. humans scoring models. This would potentially be very useful for comp social science work that attempts to analyse this more and understand the difference.
7. Section 5.2.2: can we see a table of examples of LLMs winning/preferring answers to be able to characterise why this happens? The numbers in the table don’t seem to give intuition about this, and whether this seems to happen just for some domains or overall.
8. Also, if it is intrinsically related to diversity of the text, then that would be good to verify, however if there is some other reason e.g., the rater pool/quality resulting in lower scores, it would be good to sanity check and verify this with examples.
9. Section 5.2.3: when talking about subtle shifts in the language of models, can we measure this empirically and see results over a test set? There are lots of metrics from work on style transfer (e.g., measuring perplexity shift, using classifiers for sentiment/style/tone detection and so on) that can be used here. This could help verify if there is really something measurable in the differences observed in the models or it is just a qualitative observation over some samples.

**Reasons To Accept:**

1. This paper is very well executed and explained in a good amount of detail.
2. It is empirically sound and the metrics chosen make sense (however see questions in detail below for some additional evaluations that would be good to have).
3. The authors make sure to use most of the new/large language models that are publicly available so it is good to see a large range of comparison
4. ^in light of this, the results are actually very intriguing in that different models seem to perform differently in their preferences of types of data/the ai-washing effects. This potentially hints at a way of uncovering differences in input/training of the model by this sort of post-hoc analysis of model generations when evaluated for the same effect (e.g., ai washing or exacerbation of a certain type of tone or bias and so on).
5. The appendix contains a good amount of detail and qualitative examples of model generations which is helpful to verify against the empirical results.
6. The authors introduce a term “model autophagy” and clearly define it before using it throughout the paper which is helpful!

**Reasons To Reject:**

None, but see questions and modifications below.

---

> ### Author Rebuttal · Authors · 2024-05-31
>
> Thanks for your valuable feedback. Our response is as follows:
>
> 1. **citations**
>     * We will fix related work to add the original papers for the core concepts.
>
> 2. **Cross-Scoring**
>     * For cross-scoring, the LLM functions both as a question generator (Appendix B.1) and evaluator (Appendix B.3). Each LLM scores its own answer, the answers of other LLMs and human answers, exhaustively over all pairs.
>
> 3. **Sec 4.2**
>     * We will rephrase this sentence.
>
> 4. **Sec 5.2.1**
>     * We will add following insights preferably guided by theoretical justifications.
>
>         - Why LLMs exhibit style preferences, leading an autorater to favor its own generation guided by perplexity and alignment with its training data.
>
>         - During pre-training, why the model's objective having learnt from diverse datasets leads to higher perplexity for data generated by other models.
>
> 5. **Human annotators**
>
>     * Our observation is as follows:
>
>         - **Lower Diversity in AI-Generated Text** Text generated by AI models is less diverse than that of humans, leading to relatively consistent human judgments of AI-generated content. The lack of diversity is largely due to the alignment mechanisms, like RLHF, DPO, etc.
>
>         - In contrast, human-generated texts are inherently more diverse, resulting in highly variable human judgments, as the alignment between individual humans is lower than that between large models and humans.
>
> 6. **Table 2**
>     * We will provide the score distributions for humans scoring humans and humans scoring models reflecting the diversity in style and quality.
>
> 7. **Sec 5.2.2**
>     * We will add a table of examples for LLMs winning/preferring answers to let readers characterize their findings.
>
> 8. **Diversity**
>     * Our findings indicate that the crowdsourced annotators are influenced by text diversity and style when evaluating, also supported by Wu et al.[2]. Yet, as mentioned before, we will add examples to let the readers judge and evaluate their findings.
>
> 9. **Sec 5.2.3**
>     * In our experiments, we used cosine similarity to measure these subtle changes, demonstrating that such changes lead to a decrease in text diversity. In future, we aim use mechanisms inspired from detect LLM-generated texts for a detailed quantitative analysis.
>
> **Reference**
>
> [1] Li et al. Self-Alignment with Instruction Backtranslation. ICLR, 2023.
>
> [2] Wu & Aji. Style Over Substance: Evaluation Biases for Large Language Models. ArXiv, 2023.

---

> > ### Comment · Reviewer_tbTD · 2024-06-05
> > **Thanks for the reply!**
> >
> > I will leave my (positive) score unchanged and think the paper will be greatly enhanced if the authors follow through with the above revisions.

---

### Official Review · Reviewer_rYNb · 2024-05-11

**Rating:** 7
**Confidence:** 4
**Ethics Flag:** 1

**Summary:**

This paper analyzes the issues of:
1. LLMs being used to generate more and more data followed by next versions of LLMs being trained on this synthetic data
2. LLMs being used as annotators
3. LLMs being used as the generators and the selectors of their training data.
The authors terms these phenomenon as autophagy. They analyze self-consumption loops from LLM perspective. This is done via two kinds of studies:
a. Cross-scoring: whether LLMs and humans remain impartial while filtering and transmitting information and, if not, what kind of bias they induce. This is done on a question-answer dataset. They use human annotators for generating/scoring from human perspective.
b. AI-washing: explore the risks posed by large models and humans as information generators, and to observe the changes in real data after multiple rounds of AI refinements. This is done on a text dataset and an image dataset. They analyze things such as diversity loss after N iterations.

**Questions To Authors:**

1.  Was cosine similarity computed on Bag of words? or TF-IDF? it is not fully clarified

**Reasons To Accept:**

1. Their findings on self consumption for cross-scoring are quite interesting such as both humans and LLMs showing preferences for synthetic data over human generated data
2. The experimental setup is explained well and the analysis is thorough. They have tried various prompts to prompt the LLMs into being better or worse judges for the cross-scoring experiment.
3. The appendix is quite thorough and explains many parts of their experimental process.
4. The general problem of self-consumption pointed out and analyzed by the authors in this work is an important one.

**Reasons To Reject:**

1. The paper is hard to follow at times. There is some unnecessary verbiage and complicated sentence structure.
2. This paper is more of a study/analysis. It is very well motivated, but naming it MONAL makes it seems like it's a new algorithm/model.

---

> ### Author Rebuttal · Authors · 2024-05-31
>
> Thanks for your valuable feedback. It will help a lot in improving the overall content of the paper. Our response for each individual point is as follows:
>
> **Q1. Complicated sentence structure:**
>
>  * Thank you for your suggestion. We will re-iterate the entire paper thoroughly to improve the readability of the content presented in the paper.
>
> **Q2. Monal as Name:**
>
> * In light of your feedback, we are considering renaming our paper to better reflect its content and purpose. We plan to update the name "Monal" to "a systematic framework for LLM-vs-human interaction analysis" to accurately convey the study's focus.
>
> **Q3. Cosine Similarity:**
>
>  * As mentioned in Section 5.1, we selected the [bge-large model](https://huggingface.co/BAAI/bge-large-en) as our embedding model. In Appendix C.2, we provided a detailed description of our cosine similarity computation method. Specifically, we computed the cosine similarity of the normalized text embedding vectors.

---

> > ### Comment · Reviewer_rYNb · 2024-06-03
> >
> > Thank you for your response and clarifications. I think another reviewer mistakenly responded to this.
> > I want to retain my original rating at this time.

---

### Official Review · Reviewer_xwWW · 2024-05-12

**Rating:** 5
**Confidence:** 3
**Ethics Flag:** 1

**Summary:**

The paper studies the impact of having AI in the loop of producing new AI models. It is related to the topic of synthetic (model-generated) data for training (both deliberately and not deliberately since AI-generated contents are all over the web), which has become much more popular since the release of ChatGPT. The authors frame such process as “Autophagous Loops”.

In this background, the authors propose a framework called MONAL (Model Autophagy Analysis) for understanding the interaction of LLM-generated content and human society. Through experimenting with cross-scoring, exam scenario simulation, and AI-washing, the authors analyze how LLMs and humans evaluate and filter information. The findings highlight the biases exhibited by LLMs in information processing and raise concerns about potential threats to information diversity in the age of synthetic data.

Overall I think the paper touches upon an important and challenging research topic. However, the research questions of the study can be more clearly defined and the conclusions should be more rigorous and grounded in the experiments being done.

**Reasons To Accept:**

This paper addresses a crucial topic: the increasing prevalence of model-generated content in training data and its potential influence on future AI models. With the explosion of AI-generated content online since the release of ChatGPT, understanding the implications of this "Autophagous Loop" is crucial for the responsible development and deployment of AI.

**Reasons To Reject:**

1. The paper's focus on a limited set of LLMs, primarily ChatGPT, hampers its ability to make generalized conclusions about LLM behavior. The quality and characteristics of the content generated by models, which are likely influenced by model architecture and training data, should be more carefully considered. Different LLMs can exhibit different behaviors in the tested scenarios, and this nuance cannot be easily summarized as a simple comparison of "large models vs. humans." Expanding the experimental scope to include more diverse models would significantly strengthen the paper's conclusions.

2. While the paper's framing of "Autophagous Loops" is interesting, it remains somewhat abstract and loosely connected to the specific experiments conducted. The two experimental setups explored in the paper, while illustrative, do not fully represent the vast and complex landscape of human-LLM interaction (specifically, much of the content, e.g. Section 3 and 4.1, do not tightly connect to the experiments, which are focusing on 3 specific setups). Focusing the discussion more directly on the findings of the three core experiments and their specific implications within the "Autophagous Loop" framework would make the paper's argument more convincing.

---

> ### Author Rebuttal · Authors · 2024-05-31
>
> Thanks for your valuable feedback. Our response for each individual point is as follows:
>
> **Q1. Diverse LLMs**
>
> * We argue we paid significant importance to the diversity of LLMs for evaluation. We chose six different models (Table 1) covering a wide variety of model families. E.g., we used (i) the OpenAI family, i.e., ChatGPT and GPT-4; (ii) Google's PaLM; (iii) Anthropic's Claude; and (iv) the LLaMA family, i.e., LLaMA 2-70B and SOLAR-0-70B (fine-tuned from LLaMA 2-70B). It would be great if you could point out a specific LLM or a model family, and we would be happy to add its experiments to the final draft.
>
> * In Section 5.2.1, we also analyze results not only by distinguishing between "humans and LLMs" but also by examining the behaviors of different model families. E.g., certain models tend to favor responses from their own families. Note, our results are aligned with recent studies [1], and [2] that used a similar set of model families. We will include these references in our paper to signify our model selection and strengthen our discussion.
>
> **Q2. LLMs-Human Interactions**
> * Sec 3 and 4
>   - To re-emphasize, Section 3 of the paper provides the theoretical foundation and conceptual framework necessary to understand the subsequent experimental setups.
>   - Also Section 4.1 introduces the core components of the model involved in modeling human-AI interactions, i.e., data loops, humans and LLMs. Finally, Section 4.2 builds upon 4.1 to introduce three different experimental setups.
>
> * Human-AI Interaction
>    - The landscape of Human-AI interaction is very complex and hard to model. We narrowed down this complex phenomenon to only three experimental setups focused on a framework relying on "Autophagous Loops".
>
> * Core Findings
>     - In light of your suggestion we will further inter-connect the findings of the three core experiments and their implications within the "Autophagous Loop" framework. For instance, we will further elaborate on how biases in human and language model evaluation ( in 5.1 and 5.2) may lead to synthetic data propagation, green lines in the framework of Figures 1-2. Additionally, we will emphasize how the iterative use of generative models in 5.3 leads to a decrease in information transmission.
>
> **Reference**
>
> [1] Wu et al., Style over substance: Evaluation biases for large language models. Arxiv, 2023
>
> [2] Panickssery et al., Llm evaluators recognize and favor their own generations. ArXiv, 2024

---

> > ### Author Response · Authors · 2024-06-04
> > **Rebuttal Response**
> >
> > Hi Reviewer,
> >
> > Thanks for your feedback. Please let us know if we were able to successfuly answer your questions and whether it helps you re-evaluate our scores..? Also if you have some additional questions..?
> >
> > Thanks

---

> > ### Author Response · Authors · 2024-06-06
> >
> > Hi Reviewer,
> >
> > Thanks for taking the time to review our work and response. However, we think our current scores are not truly indicative of whether your concerns have been addressed or not. It will be helpful if you can highlight some key issues in the paper that you think MUST be addressed or reconsider the score of our paper...?
> >
> > Thanks for your time and efforts!

---

### Decision · Program_Chairs · 2024-07-10

**Decision:**

Accept

**Comment:**

This paper is well-written and proposes an interesting new concept of “model autophagy”, which is important to build upon in future research. It is also empirically well-motivated, and the evaluation procedure used is sound and relevant, and based on a number of different LLMs tested, with additional details provided in the appendices.

As pointed by one of the reviewers, the paper could be improved by more clearly defining the research questions, as well as making the conclusions more rigorous and grounded in the context of the experiments being carried out. Reviewer tbTD also has a series of questions and points that should be specifically addressed in the camera-ready version of the paper. Reviewer vpdp also raises some (more minor) issues that should be changed in the final version of the paper to make the results more compelling and clear.

[At least one review was discounted during the decision process due to quality]